# Recellularized Colorectal Cancer Patient-Derived Scaffolds as In Vitro Pre-Clinical 3D Model for Drug Screening

**DOI:** 10.3390/cancers12030681

**Published:** 2020-03-13

**Authors:** Francesca Sensi, Edoardo D’Angelo, Martina Piccoli, Piero Pavan, Francesca Mastrotto, Paolo Caliceti, Andrea Biccari, Diana Corallo, Luca Urbani, Matteo Fassan, Gaya Spolverato, Pietro Riello, Salvatore Pucciarelli, Marco Agostini

**Affiliations:** 1Fondazione Istituto di Ricerca Pediatrica Città della Speranza, 35129 Padua, Italy; francesca.sensi@unive.it (F.S.); m.piccoli@irpcds.org (M.P.); d.corallo@irpcds.org (D.C.); 2Department of Molecular Sciences and Nanosystems, Ca’ Foscari University of Venice, 30172 Mestre (Venice), Italy; riellop@unive.it; 3First Surgical Clinic, Department of Surgery, Oncology and Gastroenterology, University of Padua, 35128 Padua, Italy; e.dangelo@irpcds.org (E.D.); gaya.spolverato@unipd.it (G.S.); puc@unipd.it (S.P.); 4LIFELAB Program, Consorzio per la Ricerca Sanitaria-CORIS, Veneto Region, 129 Padua, Italy; andrea.biccari@unipd.it; 5Department of Industrial Engineering, University of Padua, 35131 Padua, Italy; piero.pavan@unipd.it; 6Department of Pharmaceutical and Pharmacological Sciences, University of Padua, 35131 Padua, Italy; francesca.mastrotto@unipd.it (F.M.); paolo.caliceti@unipd.it (P.C.); 7Institute of Hepatology, Foundation for Liver Research, London SE5 9NT, UK; luca.urbani@researchinliver.org.uk; 8Faculty of Life Sciences & Medicine, King’s College London, London WC2R 2LS, UK; 9Surgical Pathology Unit, Department of Medicine, University of Padua, 35100 Padua, Italy; matteo.fassan@unipd.it

**Keywords:** 3D culture model, colorectal cancer, extracellular matrix, decellularization, drug test, response to treatment

## Abstract

Colorectal cancer (CRC) shows highly ineffective therapeutic management. An urgent unmet need is the random assignment to adjuvant chemotherapy of high-risk stage II and stage III CRC patients without any predictive factor of efficacy. In the field of drug discovery, a critical step is the preclinical evaluation of drug cytotoxicity, efficacy, and efficiency. We proposed a patient-derived 3D preclinical model for drug evaluation that could mimic in vitro the patient’s disease. Surgically resected CRC tissue and adjacent healthy colon mucosa were decellularized by a detergent-enzymatic treatment. Scaffolds were recellularized with HT29 and HCT116 cells. Qualitative and quantitative characterization of matched recellularized samples were evaluated through histology, immunofluorescences, scanning electron microscopy, and DNA amount quantification. A chemosensitivity test was performed using an increasing concentration of 5-fluorouracil (5FU). In vivo studies were carried out using zebrafish (*Danio rerio*) animal model. Permeability test and drug absorption were also determined. The decellularization protocol allowed the preservation of the original structure and ultrastructure. Five days after recellularization with HT29 and HCT116 cell lines, the 3D CRC model exhibited reduced sensitivity to 5FU treatments compared with conventional 2D cultures. Calculated the half maximal inhibitory concentration (IC_50_) for HT29 treated with 5FU resulted in 11.5 µM in 3D and 1.3 µM in 2D, and for HCT116, 9.87 µM in 3D and 1.7 µM in 2D. In xenograft experiments, HT29 extravasation was detected after 4 days post-injection, and we obtained a 5FU IC_50_ fully comparable to that observed in the 3D CRC model. Using confocal microscopy, we demonstrated that the drug diffused through the repopulated 3D CRC scaffolds and co-localized with the cell nuclei. The bioengineered CRC 3D model could be a reliable preclinical patient-specific platform to bridge the gap between in vitro and in vivo drug testing assays and provide effective cancer treatment.

## 1. Introduction

Globally, colorectal cancer (CRC) is one of the most common malignancies in the world, third diagnosed cancer cause and second cancer cause of death, accounting for more than 1.8 million estimated new diagnoses (crude rate 24.2 per 100,000 people) and more than 880,000 estimated deaths (11.5) in 2018 [1].

Nowadays, surgery is a curative treatment for these patients, but 40 to 50% who undergo surgery alone, ultimately relapse and die of metastatic disease [2,3,4]. In this context, it is important to know that patients with high-risk stage II and stage III colon cancer are randomly assigned to receive an adjuvant treatment in order to reduce recurrence and metastasis formation [5]. The known drug combinations in CRC treatment are FOLFIRI (leucovorin + 5-fluorouracil (5FU) + irinotecan), and FOLFOX (leucovorin + 5FU + oxaliplatin) that has been shown to improve overall survival by more than 20% compare to surgery alone and is considered a standard treatment [6,7,8]. However, the vast majority of patients will not benefit from receiving adjuvant chemotherapy because they have already been healed by surgery or because of the development of drug resistance [9].

An important point in the drug discovery process is the preclinical evaluation of drug cytotoxicity, efficacy, and efficiency. To date, immortalized cancer cells have significantly contributed to the knowledge about tumor biology, signaling pathways, and the investigation of new drug treatments [10]. Nevertheless, so-called flat cells fail to represent the real complex spatial architecture of in vivo tissues [11]. It has been estimated that only 5% of active drugs in 2D cellular models are found to be active in clinical trials [12]. For this reason, in the past decades, significant efforts have been made toward the improvement of models capable to bridge the gap between the in vitro and in vivo experimental phases. To date, the 3D patient-derived models, mostly used in the literature to perform drug evaluations, are organoids [13]. Organoids are a valid model with a significantly improved level of complexity with respect to cells grown on tissue culture plastic. At the same time, the organoid technology is still too close to the vision that associates the cancer phenomenon with a cellular-only disease. A plethora of research works demonstrate that the microenvironment is not only architecture capable of supporting the tissue but also a complex network of molecules that contribute in a fundamental way to the disease evolution [14].

Tissue extracellular matrix (ECM) function in determining cell fate and behavior has visibly emerged in the last two decades. ECM is a physiologically functional tissue element, responsible for cell adhesion, cell–cell communication, and cell proliferation [15]. Basically, the ECM is composed of glycosaminoglycans (GAGs), most often covalently linked to several proteins forming the proteoglycans, and fibrous proteins as collagen, elastin, fibronectin, and laminin [16]. These components are secreted locally by the cells and assembled into the organized meshwork that is the ECM microenvironment. Every organ has a distinctive ECM composition to serve particular tissue-specific purposes and functions. This exclusive combination of factors arises from dynamic biological/physical/chemical interactions among cellular components and their microenvironment during tissue development and disease progression [17]. Given the increasingly prominent role that ECM has achieved, a lot of studies have moved to address cancer not only as a disease of uncontrolled cell proliferation but also of dysregulation of the tissue-specific microenvironment.

In this study, we combined tumor cells with CRC ECM in order to obtain a 3D construct, which could be easily manipulated in the laboratory and that simulated pathological characteristics of the tumor. In this novel in vitro CRC model, we evaluated the activity of specific drugs with the aim to translate it as a preclinical model before chemotherapy administration in patients. We recently demonstrated that the decellularized healthy tissues and neoplastic counterparts were biologically active scaffolds able to support the in vitro cell growth and proliferation [14,18]. The decellularization process enables complete cell removal, maintaining at the same time the tissue architecture and preserving the ECM components [19]. Based on this approach, we investigated the effects of a standard chemotherapy administration in the 3D CRC model obtained combining the HT29 and HCT116 adenocarcinoma cell lines and patient-derived ECM scaffolds to validate the in vitro model for drug screening and treatment outcome prediction.

## 2. Results

### 2.1. Generation and Characterization of Repopulated 3DN and 3DT Models

Biopsies of the resected primary tumor (T) and matched normal (N) healthy counterpart were obtained (Table 1) and decellularized with two detergent-enzymatic treatment (DET) cycles, as previously demonstrated [14]. In Appendix A, a gross appearance of fresh (FN and FT) and decellularized (DN and DT) biopsies underlined the typical color change from red to white. A global overview of tissue architecture was performed through H&E staining and laminin immunostaining, where we demonstrated that the ECM was maintained in both N and T samples, and cells completely disappeared. In addition, SEM analysis confirmed the ultrastructure preservation after DET cycles (Appendix A).

HT29 and HCT116 adenocarcinoma cell lines were chosen to repopulate decellularized scaffolds because they are derived from two different malignant levels and represent different extents of differentiation: HT29 cell line has an intermediate capacity to differentiate, while HCT116 cells are highly aggressive and show no ability to differentiate. Qualitative and quantitative characterization of matched normal (3DN) and tumor (3DT) HT29-recellularized samples were performed through histological, immunofluorescence, and DNA amount quantification. HT29 adenocarcinoma-derived cells cultured in decellularized scaffolds were uniformly distributed, initially occupying the outermost part, and then penetrating inside the scaffolds. Nuclei were clearly visualized with no signs of cellular suffering. Cells populating the tumor scaffolds were organized in a rounded configuration that, according to their origin and pathophysiological arrangements, is characteristic of dysplastic colic crypts (Figure 1A). The H&E staining revealed a large number of spherical cell aggregates that repopulated the porous spaces within the scaffolds (Figure 1A). These round gland-like aggregates strongly adhered to the ECM composing the scaffolds. Collagen IV staining underlined a uniform presence of collagen fibers that were equally distributed in 3DN and in 3DT tissues (Figure 1A). The periodic acid–Schiff (PAS) staining evidenced the presence of secreted and organized glycosaminoglycans and glycoproteins (Figure 1A). In addition, PAS staining showed that in both recellularized healthy and tumor tissues, HT29 injected cells actively produced and secreted mucus. SEM analysis confirmed the presence and morphology of cells within the tissues (both normal and tumor), especially near the injection site (Figure 1A). Quantification of DNA of repopulated 3D constructs, with both HT29 and HCT116 cells from healthy and tumor specimens after 5 days of culture, showed significant restoration of the DNA amount compared with decellularized scaffolds in both healthy and tumor tissues (*p*-value < 0.05) (Figure 1B and Appendix A, respectively).

### 2.2. Evaluation of Tumor Cell Proliferation: 2D vs. 3D Setting

To test the ability of decellularized scaffolds to support cell colonization and viability, we evaluated the proliferation rate of the injected HT29 cells through the proliferation marker Ki67 (Figure 1C). As represented by IF analysis, the number of Ki67-positive cells in 3DT specimens (76 ± 4.5%) was significantly higher compared with 3DN (50 ± 7.2%) at day five of culture (*p*-value = 0.0180) (Figure 1C). Furthermore, IF analysis evidenced a decreased proliferation rate in the 3D model when compared with the cells grown in conventional 2D condition (92 ± 1.4) (Figure 1C and Appendix A). This data was in accordance with the literature [20,21,22]. A similar trend was observed also in HCT116 repopulated scaffolds in which we found a significant increase in Ki67 positive cells in 3DT (69 ± 3%) compared with 3DN (54 ± 4.6%) at day 5 of culture (*p*-value = 0.0140) (Appendix A). 

Since the adhesion and cytoskeletal molecules play a vital role in cell morphology and function [23], we investigated whether the expression and localization of E-Cadherin and vimentin molecules differed among cells cultured in 3DN, 3DT, and 2D conventional culture settings. The typical epithelial-like phenotype of HT29 cells observed in conventional 2D culture condition was fully preserved in both the repopulated 3DN and 3DT, as evidenced by the high expression of the epithelial cell adhesion marker E-cadherin (97.4 ± 0.8% in 2D, 93.6 ± 5.3% in 3DN, and 91.9 ± 3.6% in 3DT) and the very low expression of the mesenchymal adhesion molecule vimentin (9.58 ± 0.63% in 2D, 3.6 ± 0.9% in 3DN, and 4 ± 0.8% in 3DT) (Figure 1C and Appendix A) [24,25]. Differently, the HCT116 cell profile, when cultured in 2D and 3D conditions, did not completely overlap. HCT116-repopulated scaffolds confirmed a similar expression of E-cadherin in both settings: high expression in 2D and high expression in 3D (97.86 ± 0.93% in 2D, 87.4 ± 5.1% in 3DN, 84.5 ± 4.1% in 3DT) but showed an increased expression of vimentin compared with both 2D culture and HT29-repopulated scaffolds, suggesting a contribution given by the microenvironment in the modulation of the cellular profile according to the growth setting condition (12.55 ± 2.39% in 2D, 36.8 ± 4.9% in 3DN, and 46.7 ± 7.5% in 3DT) (Appendix A) [26].

### 2.3. Effect of 5FU Treatment on Cells Cultured in 2D and 3D Model

5FU is a widely used drug in CRC adjuvant chemotherapy and still represents the backbone of different multimodal treatments [27]. Firstly, we calculated the percentage of viable cells at a different time point (0–72 h), treated with an increasing drug concentration (0.1 µM–100 µM) (Appendix A). As previously shown [27], we confirmed that the in vitro 5FU IC_50_, for both HT29 and HCT116 cells, corresponded to 1.3 µM and 1.7 µM, respectively (Appendix A). Secondly, we compared the efficacy of 5FU treatments in 2D conventional HT29 cell culture and 3DN and 3DT culture models. In accordance with the 2D model, the 5FU response of 3DN and 3DT models maintained a dose-dependent trend (Figure 2A). If compared to the 2D conventional culture, HT29 cells grown in both 3DN and 3DT models displayed a reduced sensitivity to 5FU (3DN vs. 2D: *p*-value < 0.001 at 1-10-100 µM; 3DT vs. 2D: *p*-value < 0.001 at 1 and 10 µM; *p*-value < 0.01 at 100 µM), with an increased IC_50_ of about 10-fold (IC_50_ = 11.58 µM) (Figure 2A,B). We also calculated IC_50_ in 2D and 3D models using FOLFIRI, and, in accordance with 5FU, IC_50_ was increased (2D IC_50_ = 3.14 µM and 3D IC_50_ = 12.54 µM) (Figure 2B). Similarly, if compared to 2D cultures, HCT116 cells grown in both 3DN and 3DT models displayed a reduced sensitivity to 5FU (3DN vs. 2D: *p*-value < 0.001 at 1-10-100 µM; 3DT vs. 2D: *p*-value < 0.001 at 10 and 100 µM; *p*-value < 0.01 at 1 µM) (Appendix A), with an IC_50_ of 9.87 µM (Appendix A). Given the similarity of results using different types of treatment (5FU and FOLFIRI) and considering that 5FU is the drug used in all the most common therapeutic combinations, for simplicity, we performed all the subsequent analyses using only this molecule. The IC_50_ values in 3DN and 3DT models were further confirmed by additional immunohistochemistry analyses, where mindbomb E3 ubiquitin protein ligase 1 (MIB1)-positive cells showed a significant decrease of about 50% in both HT29-repopulated 3DN and 3DT samples when treated with 5FU 3D IC_50_ (3DN *p*-value = 0.0089; 3DT *p*-value < 0.0001, compared to non-treated repopulated scaffolds) (Figure 2C). In order to further corroborate the different response to treatment between 3D and 2D settings, EdU and TUNEL assays were performed to detect proliferation and apoptosis, respectively. We identified a significant proliferative phenotype in non-treated (-5FU) 3DN and 3DT models (56.6 ± 3.1% and 61.5 ± 4.3% of EdU-positive cells; 2.1 ± 0.4% and 1.9 ± 0.8% of TUNEL-positive cells), a situation that was completely reversed after 5FU treatment using the calculated 3D IC_50_ (15.5 ± 4.1% and 12.7 ± 1.7% of EdU positive cells; 45.5 ± 6% and 29.6 ± 7.4% of TUNEL positive cells) (Figure 2D,E). These findings underlined the healthy behavior of HT29 and HCT116 cells cultured in 3D models and their sensibility to standard chemotherapy treatment using a specific dosage.

### 2.4. Effect of 5FU Treatment on In Vivo in a Zebrafish Model

To validate the in vitro results obtained using the 3D culture with an in vivo system, we generated a zebrafish (*Danio rerio*) xenotransplantation model. Under the microscope, fluorescently labeled human cancer HT29 cells were transplanted into transgenic zebrafish Tg (fli:EGFP) embryos in the duct of Cuvier (Doc) 48 h after fertilization (Figure 3A). Doc is the blood circulation valley on the yolk sac, connecting the heart to the trunk vasculature. In this way, cancer cells are introduced directly into the circulatory system and are able to diffuse throughout the embryonic body via the bloodstream after the injection [28]. Cells labeled with the Dil live-cell tracer rapidly distributed throughout the vasculature of the injected embryos and were maintained up to 72 h (Figure 3B). In particular, we performed our analyses on 48 h since the percentage of untreated embryos retaining the cells after 72 h post-injection was inferior to 50% (Figure 3B) and were automatically discarded from the analysis. To investigate the drug concentration sufficient to remove at least half of the injected cells, we incubated the xenotransplanted zebrafishes with 5FU directly administered into the embryo medium. In order to compare the effects previously observed in the 2D and 3D models, we used the IC_50_ of 2D and 3D cultures, respectively. We evaluated the fluorescence fold change in the caudal region of each injected embryo after 24 and 48 h of treatment. Interestingly, we observed a significant reduction of Dil-positive cells only in embryos treated with 3D IC_50_ when compared with control (DMSO-treated embryos), especially after 48 h of treatment (*p*-value = 0.0027) (Figure 3C,D). These data suggested that the response to treatment observed in the in vitro CRC 3D model recapitulated the effect registered in vivo in a xenogeneic zebrafish model.

### 2.5. 3D Tumor Permeability

To confirm that the pharmacological effect observed in our 3D model was not the result of a lack of nutrient and drug permeability, but specifically to drug activity in a 3D microenvironment, we analyzed patient-derived scaffold permeability, drug diffusion, and localization when administered to a 3D tumor construct.

In order to verify drug absorption in 3D samples with a mean thickness of 1000 µm (thickness range: from 520 µm to 1734 µm; Figure 4A), we recellularized tumor scaffolds with HT29 ZSgreen positive cells for five days and then added the doxorubucin (doxo). Doxorubicin was chosen because auto-fluorescent (emission wavelength: 594 nm) and, therefore, its localization could be detected through microscopy analysis [29]. After 72 h of treatment, we found a high percentage (65 ± 3.6%) of HT29 ZSgreen positive cells and doxo co-localized events (Figure 4B,C), and IF analyses confirmed the presence of the drug inside cell nuclei (Figure 4B). Importantly, these co-localizations were not only present on the top of the scaffolds but also in the inner core of the treated 3D constructs (Figure 4B). Despite the scaffold architecture complexity, we noticed that doxo did not bind to the ECM (Figure 4B).

Experimental results of the filtration process and relative permeability K quantifications carried out on fresh, recellularized (3DT), and decellularized tumor tissues (Figure 4E and Appendix A) showed a very interesting trend where 3DT resulted in between fresh and decellularized sample permeability. As expected, permeability in decellularized tumor samples was significantly higher compared to fresh tumors (*p*-value = 0.036) (Figure 4E,F). The increased permeability obtained through a simple in-house developed device, described in Figure 4D, confirmed the accessibility of the cells seeded in the scaffold to the drug administered through the culture media, even in the most inner areas of the construct. These data indicated that our 3D model was suitable for in vitro drug testing.

## 3. Discussion

The majority of high-risk stage II and III CRC patients do not completely benefit from receiving standard chemotherapy during adjuvant treatments, probably because they have already been cured by surgery or due to the chemotherapy resistance. For these patients, the benefit of adjuvant chemotherapy remains under debate [30,31]. Moreover, the success rate for many drugs evaluated in conventional cellular and animal models tested in clinical trials is very low [32]. Today, the random assignment to adjuvant chemotherapy without any predictive factor of efficacy is one of the critical points in the management of CRC patients. Therefore, the development of an in vitro model that could be useful for drug screenings and capable of faithfully mimicking the clinical response to specific drugs is a strong clinical need. To discover new drugs, thus trying to shorten the in vitro preclinical testing phases, researchers are evaluating the 3D approach as a way to overpass the gap between the in vitro and in vivo experimental phases.

A limit to the definition of a 3D model is the fact that many concepts are associated with this acronym. 3D, for example, is associated with multicellular spheroids, organoids, synthetic scaffolds, hydrogels, organs-on-chips, and 3D bioprinted structures, each with their own advantages and disadvantages [33]. Up to date, organoids are the most studied 3D culture, both in cancer research and regenerative medicine field [34,35]. We believe that the analysis of organoids, even if performing, is still too tied to a vision that associates the cancer disease with a cellular-only disease. Tumors are surrounded by ECM and stromal cells, and the pathophysiological state of the tumor microenvironment is closely associated with tumorigenesis. Microenvironment co-evolves into an activated state during all of the dynamic tumorigenesis processes: initiation, progression, and, finally, metastasis. Indeed, many of the hallmarks of cancer, delineated by Hanahan and Weinberg, are provided by various stromal components, including ECM, endothelial cells, fibroblasts, leukocytes, and so on [36]. In this context, growing evidence indicates that ECM is not only physical support for the cells but also an active component of tumor tissues, implicated also in response to therapy [37]. For this reason, we focused our attention on the generation of a reliable in vitro 3D preclinical model useful for drug evaluation without the introduction of any synthetic component. Thanks to the regenerative medicine principles, our goal was to re-create in vitro a model of the pathological tissue compatible with the patients of origin. In this study, CRC patient-derived ECM served as a patient-specific microenvironment and was used for evaluating its role (1) as support for tumor cell growth and proliferation and (2) as a patient surrogate platform for chemo-sensitivity analysis in 3D conditions. Finally, to validate 3D patient-derived CRC, we compared the 5FU drug effect in the conventional 2D culture model and a xenotransplanted zebrafish in vivo model.

Initially, our work was focused on the analysis of both healthy and pathological tissues derived from the same patient. In particular, we performed recellularization and characterization experiments in order to investigate if both types of tissues were able to sustain cell maintenance and proliferation, and we confirmed this aspect. Later, when we focused on the assessment of chemosensitivity, and in the comparison of drug response in different biological models as a final goal, we decided to focus our attention on the tumor setting alone since HT29 or HCT116 repopulated healthy scaffolds are chimeric constructs, which do not reflect the pathophysiological conditions of a tumor nor a healthy colonic tissue. In other words, with the idea to replicate in vitro a patient’s tumor biopsy, only tumor-derived scaffolds and cancer cells were taken into account in the chemosensitivity assays and characterization. A decellularized protocol, recently reported by our team [14], combines the use of sodium deoxycholate and DNase to obtain acellular ECM-scaffolds to support cell survival and model their behavior. This methodology was applied to healthy human colon mucosa and matched CRC biopsies from the same patient, revealing that the collagens and glycosaminoglycans were properly maintained and distributed within the scaffolds, also after the DET process [14]. The importance of the ECM components in maintaining the tissue homeostasis is exemplified by the study of Weaver et al., in which they reverted the malignant phenotypes of breast cancer cells to the wild-type phenotype [38]. This was demonstrated by culturing breast cancer cells onto basement membrane-based 3D substrates coated with integrin β1-blocking antibodies. This study confirmed that the crosstalk between the ECM and the populating cells could reverse the cancer phenotype, emphasizing the ECM role in modulating cell behavior and phenotype. In addition, the recent models proposed by Tian et al. [39] and Girdhari Rijal et al. [40] were all valid, interesting, and finely evaluated different approaches, capable of implementing the research of cancer development and progression. They generated constructs from materials that were fabricated using native mouse-tissue ECM, which were processed and used as a coating material for monolayer or multilayer tissue cultures. We believed that these models certainly offered great progress compared to simple 2D monolayer cell culture, but did not consider the complexity of a three-dimensional tissue structure where the biological and topological properties of the original organ-specific ECM play pivotal roles. To evaluate the ability of the ECM, in sustaining cancer cell proliferation, we used HT29 and HCT116 colorectal adenocarcinoma stabilized cells. We observed that the process of recellularization was reproducible and reliable, and the different cells seemed to colonize the scaffold to a similar extent. In the 3DT model, we observed that tumor cells were located inside the crypts and on the top of the luminal area. Moreover, immunohistochemistry analyses revealed that tumor cells acquired different phenotypes based on their location in the ECM. The cells inside the crypts were more rounded and organized to leave a lumen in the center; instead, the cells that were placed on the top of the luminal area were thinner in order to create a multi-layer structure following the anatomical morphology of colon cells. We observed that cells in the outer layer were in active proliferation probably because they had access to nutrients and oxygen, simulating the in vivo vascularized region of the tumor zone close to the capillaries [41].

The 3D ECM-scaffolds sustained HT29 cell differentiation, leading to the formation of glandular-like structures. Moreover, the presence of mucus, underlined by the intense purple staining in close proximity to the tumor cells inside the ECM, was a marker of functional differentiation of epithelial cells injected into the ECM since the cells grown alone in the standard 2D Petri dishes are known to have no mucus-secreting properties [42]. These findings emphasized the capability of the ECM components to sustain the differentiation phenotype of HT29 cells and to mimic the original tissue organization.

Interestingly, HT29 cells used to recellularized the scaffolds maintained the expected “epithelial phenotype”, showing a high expression of E-cadherin coupled with a low expression of vimentin. Differently, HCT116 was positive for E-cadherin in both 2D and 3D conditions, but vimentin expression indicated a more mesenchymal-like pattern when cultured in three-dimensional cultures. A similar result was observed by Matsuda and colleagues who analyzed epithelial–mesenchymal transition (EMT) markers in 16 colorectal cancer cell lines, including HT29 and HCT116, after orthotopic implantation into nude mice and in vitro cultures [26]. They defined “EMT phenotype” that we, also, observed for HCT116 cells, with an increased expression of vimentin in the tumors in vivo compared with a no-expression in 2D cultured cells. This cell behavior supported the idea that the HCT116 cell line acquired the “EMT phenotype” following orthotopic implantation in Matsuda study, and following 3D scaffold recellularization in our experimental setting. This data brought the 3D model closer to an in vivo model, supporting the idea that a patient-derived ECM does not result as an inert three-dimensional scaffold but a biologically active environment.

In order to understand if the contribution of the 3D microenvironment could influence the response to chemotherapy treatment, we calculated the IC_50_ in 2D conditions and in the 3D models. We demonstrated that the 3D CRC model exhibited reduced sensitivity to 5FU and FOLFIRI treatments compared with 2D conventional cell cultures. The fact that numerous anticancer drugs are subsequently discarded during the clinical evaluations indicated that the cytotoxic activity tended to be overestimated on a 2D-culture-based screening platform [43]. To compare the findings obtained with the 3D model with an in vivo system, we successfully generated a xenotransplanted zebrafish model. We considered several advantages using the zebrafish embryos: the short generation time, the fast and external embryonic development, the large number of off-spring, and, most importantly, the transparency of the embryos, which enables live and noninvasive fluorescent imaging [44]. In our experiments, we observed that the xenotransplanted zebrafish in vivo model provided 5FU IC_50_ completely comparable with the 3D model, validating what observed in our 3D patient-derived constructs. 

The differential drug sensitivity observed between 2D and 3D models might be attributed to the decreased factors access or reduced drug sensitivity in response to hypoxic and more slowly cycling cells under 3D conditions [45,46,47]. However, the specific signaling mechanisms implicated in drug resistance in such patient-derived and bioactive 3D platforms are still unclear. Saraswathy and Gong’s study showed that the interaction among various factors (e.g., intracellular changes, paracrine signaling, modification in the supporting matrix) possibly would contribute to the reduced drug sensitivity in 3D models. In particular, the hypoxia in cancer is known to lead resistance via different pathways, such as the loss of p53-mediated apoptosis and the enhanced P-glycoprotein expression [48]. To understand if our 3D model could influence gas exchange or drug absorption, we decided to evaluate the permeability of the decellularized and recellularized tissues compared to fresh specimens. As reported by the literature, we showed that the sample permeability increased in the tumoral tissues compared to the healthy counterpart, and it is interesting to note how this trend is maintained not only in the fresh tissues but also in decellularized tissues [49]. Permeability depends on many factors, such as pore size, matrix composition, and matrix geometry; we estimated the increased permeability of decellularized tumor samples in respect to the original fresh tissues in order to understand how easily these scaffolds could allow the uptake of both key factors for cell survival (e.g., nutrients, oxygen) and exogenous factors, such as drugs. The increased permeability indicated that when repopulated by the cells, our 3D model could be effectively perfused by drugs and could, therefore, be a useful preclinical model for the study of drug delivery. Indeed, when natural autofluorescent doxorubicin was used in our 3D tumor model, the assay demonstrated the co-localization of the chemotherapy drug with the cells, even in the deeper parts of the repopulated scaffolds. These data indicated that the CRC patient-derived 3D model allowed perfusion and diffusion of the drug through the scaffold, suggesting that the observed cells apoptosis was specifically linked to the pharmaceutical effect and not to aspecific cellular stress.

We are aware that our 3D model is still embryonal and still does not reach the complexity and the maturity of fresh tissue. This is evident from the general characterization, for example, the amount of DNA detected in the recellularized samples never reaches the levels of fresh tissues, but also because in this experimental phase, we added only and exclusively tumor cell lines. To talk about a mature construct, we need to consider other important players, such as stromal or immune cells, that are known to play a key role in cancer growth and progression [50]. Future analyses will be focused to test the efficacy of multi-drug treatments and how the analyzed parameters may be modified by enriched recellularization with essential CRC microenvironment components, such as tumor-infiltrating lymphocyte and cancer-associated fibroblast, that are shown to promote chemotherapy resistance by enhancing cell stemness and epithelial-mesenchymal transition mechanisms [51]. To address these topics and to improve the 3D patient-derived culture model, experiments inflow and dynamic systems need to be added to a present culture setting.

## 4. Materials and Methods

### 4.1. Patients

A total of 23 paired normal mucosa (N) and tumor lesion (T) from CRC patients who underwent curative surgery were collected from the First Surgery clinic, University of Padua (Department of Surgery, Oncology, and Gastroenterology) and General Surgery Unit, S. Antonio Hospital (Padova). All of the patients enrolled satisfied the following criteria: histologically confirmed primary adenocarcinoma of the colon, age > 18 years, and written informed consent (prot. 448/2002). Patients with a known history of a hereditary colorectal cancer syndrome and who underwent neoadjuvant treatments were excluded. This study was conducted according to the Declaration of Helsinki principles; written informed consent was obtained from the individual patients, and the ethics committees of institutions approved the protocol (Azienda Ospedaliera di Padova Ethical Committee Approved Protocol Number: P448).

### 4.2. Cell Cultures

Human colon adenocarcinoma cell lines HT29 and HCT116 were grown in Roswell Park Memorial Institute (RPMI) 1640 Medium (EuroClone, Milan, Italy) and Eagle/Dulbecco modified Eagle medium (DMEM) (PanEko, Moscow, Russia), respectively, supplemented with 10% fetal bovine serum (FBS) (HyClone, Pittsburgh, USA), 1 mM glutamine, 10 mg/mL penicillin, and 10 mg/mL streptomycin at 37 °C in a 5% CO_2_ humidified atmosphere. The lentiviral vector pHIV-Luc-ZsGreen was a gift from Dr. Bryan Welm (Department of Surgery, University of Utah, purchased through Addgene Inc. MA, USA, Plasmid #39196) and was used to generate lentivirus expressing both ZsGreen fluorescent protein and firefly luciferase via an internal ribosome entry site under the EF1-alpha promoter.

### 4.3. Tissue Decellularization

All mucosa specimens encompassed the luminal surface, mucosa, and submucosa. CRC tissue was obtained at the edge of infiltrating neoplasia; healthy colon mucosa was obtained more than 10 cm away from the CRC. After surgery, N and T were kept in sterile phosphate-buffered saline (PBS) for a maximum of 2 h before processing. All the steps of decellularization were performed with sterile solutions under tissue culture hood. N and T tissues destined to be used as fresh were rinsed with sterile PBS and consequently treated according to the methodology with which were analyzed. Normal mucosa and CRC destined to decellularization process were treated with two detergent-enzymatic treatment (DET) cycles. Each DET cycle was composed of deionized water at 4 °C for 24 h (h), 4% sodium deoxycholate (Sigma, Milan, Italy) at room temperature (RT) for 4 h, and 2000 kU DNase-I (Sigma) in 1 M NaCl (Sigma) at RT for 3 h, after washing in water. After decellularization, matrices were rinsed in 3% penicillin/streptomycin (pen/strep)/PBS for at least 4 days and then preserved at −80 °C.

### 4.4. DNA Isolation and Quantification

To assess the total DNA content within the fresh normal and tumor samples, the recellularized 3D normal tissues (3DN), the recellularized 3D tumor tissues (3DT), and the corresponding decellularized matrices, each specimen was treated using the DNeasy Blood&Tissue kit (Qiagen, Hilden, Germany) under manufacturer’s instruction. DNA quantification was performed using Nanodrop 2000 spectrophotometer at the 260/280 nm ratio (ThermoFisher Scientific, Waltham, MA, USA).

### 4.5. Immunohistochemistry and Immunofluorescence 

Frozen sections (7–8 µm thick) were stained with hematoxylin and eosin (H&E; Bio Optica, Milan, Italy), periodic acid-Schiff (PAS; Bio Optica, Milan, Italy), anti-collagen IV (1:100, Dako, Milan, Italy), and anti-MIB1 antibody (1:50, Dako, Milan, Italy). All the stainings were performed according to the manufacturer’s instructions. Formalin-fixed paraffin-embedded (FFPE) tissue specimens were used to perform immunohistochemical (IHC) staining. For immunofluorescence (IF) analysis, the sections were permeabilized with 0.5% Triton X-100, blocked with 10% horse serum and incubated with the primary antibodies laminin (1:100, L-9393 Sigma, Italy); Ki-67 (1:100, ab15580 Abcam, Cambridge, UK); E-cadherin (1:250, BD Biosciences, Dublin, Ireland), vimentin (1:100 ab92547 Abcam, Cambridge, UK), and TUNEL and EdU (Thermo Fisher Scientific, Waltham, MA, USA). Slides were then washed and incubated with the labeled Alexa Fluor secondary antibodies diluted 1:200. Finally, nuclei were counterstained with fluorescent mounting medium (Sigma-Aldrich, Milan, Italy), containing 100 ng/mL DAPI (Sigma-Aldrich). For each specimen, random pictures were collected with a direct microscope.

### 4.6. Recellularization of Scaffolds

Normal and tumor decellularized matrices were incubated overnight with growth medium containing primocin antibiotic (InvivoGen, Kampenhout, Belgium) at 4 °C. In order to normalize the intra-sample variability, scaffolds were cut into comparable dimensions before seeding. All matrices were then injected with 2.5x10^5^ cells, resuspended in 10 μL of collagen I (diluted 2:3 with RPMI-1640), using a 30G syringe needle. Samples were incubated for 4 h in a humidified incubator at 37 °C and 5% CO_2_. The complete medium was carefully added and changed every two days. Recellularized samples were either formalin-fixed and paraffin-embedded for the IHC stainings or fixed in 4% paraformaldehyde (PFA) and then included in OCT (optimal cutting temperature compound) for the IF analysis.

### 4.7. Scanning Electron Microscopy Analysis (SEM)

Fresh, decellularized, and recellularized tissues were slice into segments of approximately 0.5 cm and fixed with 3% glutaraldehyde in 0.1 M phosphate buffer. After fixation and PBS washes, the samples were dehydrated in a graded ethanol-water series from 15% to 100% ethanol, critical point dried using CO_2_, and mounted on aluminum stubs using sticky carbon taps. Samples were mounted and coated with a thin layer of Au/Pd (approximately 2nm thick) using a Gatan ion beam coater. Images were recorded with a JEOL JSM 6490 scanning electron microscope.

### 4.8. Drug Treatment and Cytotoxicity Assay

For the 5FU and FOLFIRI treatments in 3D setting, N and T scaffolds were repopulated with 2.5 × 10^5^ HT29 and HCT116 cells in 24-well plates. The FOLFIRI chemotherapy cocktail ratio used was as follows: 5FU:leucovorin:irinotecan, 25:5:1 [52]. Five days post-seeding, the cells were treated with 1 µM, 10 µM, and 100 µM 5FU for 72 h. For the 5FU treatment of 2D cultures, HT29 and HCT116 were seeded at 5 × 10^3^ cells per well in 96-well plates and treated with different concentrations of 5FU from 0.1 µM to 100 µM for 72 h. Cell viability was determined 24 h, 48 h, and 72 h post-treatment by reading the absorbance using the Multilabel Plate Reader VICTOR (PerkinElmer, Waltham, MA, USA). The treatment response for each culture setting was standardized to the corresponding untreated cultures. Similarly, the PrestoBlue cell viability reagent was used for the determination of inhibitory concentration 50% (IC_50_) using GraphPad Prism Software (version 6, GraphPad Software, San Diego, CA, USA).

### 4.9. Analysis of Cells Proliferation Rate

The 5-ethynyl-2’-deoxyuridine (EDU)–Click kit was used for the evaluation of DNA synthesis following the manufacturer’s instructions. The cells were pulsed with EdU for 4 h before fixation in 4% PFA and subsequent EdU detection. Nuclei were counterstained with DAPI (Sigma-Aldrich). EdU-positive cells were counted under an inverted fluorescence microscope (BMI6000B Leica) and normalized to the total number of nuclei.

### 4.10. Fluorescent Cell Labeling, Zebrafish Embryos Preparation, Tumor Cell Implantation, and 5FU Treatment

The *Tg(fli1: EGFP)* zebrafish embryos [53] were raised, staged, and maintained, as previously described [54]. The two days old embryos were anesthetized with 0.003% tricaine (Sigma-Aldrich) and placed on a 10 cm Petri dish containing 3% agarose. Non-fluorescent HT29 cells were labeled with the Vybrant^®^ DiI cell-labeling solution (Invitrogen, Carlsbad, CA, USA), according to the manufacturer’s instructions. The cells were resuspended in PBS, and approximately 200 cells were implanted within the duct of Cuvier of each embryo using a pneumatic picopump equipped with borosilicate glass capillary needles (OD/ID: 1.00/0.75 mm, WPI, USA). After the injection procedure, the embryos were raised at 33 °C, and, after 2 h, the animals showing less than 100 cells were discarded from the analysis. At least 50 embryos per group were analyzed from three independent experiments. Embryos were live photographed using a BM6000 (Leica, Milan, Italy) microscope equipped with a PerkinElmer UltraVIEW VoX Confocal System. In order to compare the effects observed in the 2D and 3D models to the zebrafish xenograft embryos, we used the same IC_50_ concentrations used in vitro under 2D and 3D culture conditions, respectively. 5FU was dissolved in DMSO and diluted into embryo medium to a final concentration of 1.3 μM and 11.58 μM, respectively. Forty-eight hours post-fertilization (hpf), embryos were incubated with this compound concentration, and each animal was imaged with a Zeiss Axio. Observation with a microscope for live-cell imaging was performed every 24 h. Embryos incubated with DMSO only were used as control.

The Fondazione Istituto di Ricerca Pediatrica Città della Speranza is allowed to use rodents and zebrafish for biomedical research by the Ministry of Health with Ministerial Decree No. 21/2019-UT dated 4 July 2019 that replaces the previous Ministerial Decree No. 09/2014-UT dated 30 December 2014 according to D.lgs 26/2014, concerning the protection of animals used for scientific purposes. The Institute has an internal Ethics Committee “Organismo Preposto al Benessere Animale” (OPBA).

### 4.11. Permeability Tissues Evaluation

In the permeability measure method, both biological and synthetic specimens were evaluated. A device was adopted for the tissue permeability measurement (Figure 4D). Tissue samples were confined in cylindrical space with a diameter of 3 mm and a thickness of 2 mm. Two porous metal plates were placed in the upper and lower surface to fix the samples and allow the fluid filtration. The upper metal plate was connected to a pipette with an internal diameter of 6 mm and a total height of 250 mm. The permeability of a sample was estimated, filling the pipette with fluid at an initial height of 210 mm, with respect to the inferior surface of the specimen. The lower surface of the sample was subjected to atmospheric pressure, while the upper surface to variable relative pressure, depending on the height of the fluid column. In this condition, it can be assumed that the filtration in the specimen is governed by Darcy’s Law:(1)$$Q=KA_{s}\frac{\gamma H\left(t\right)}{\Delta x}$$
where *Q* is the flux of the fluid, *K* the coefficient of permeability of the tissue to the permeating fluid, *A_s_* the transversal section of the specimen, *γ* the specific weight of the fluid, *H(t)* the height of fluid column over time t, and *Δx* the thickness of the specimen. Defined with *A_p_* the transversal section of the pipette and considering the continuity equation between pipette and sample, the Darcy’s Law can be integrated over time and rewritten as follow:(2)$$H\left(t\right)=H\left(t_{0}\right)\exp\left[-K\frac{A_{s}}{A_{p}}\frac{\gamma }{\Delta x}\left(t-t_{0}\right)\right]$$
where *t* is the current time, and *t_0_* the initial time of the filtration process.

In the experiments, the fluid column height was acquired over time. In all the experiments, this height was included in the range of 210–100 mm, corresponding to a pressure gradient in the specimen of 1.03–0.49 kPa/mm. Fluid column height vs. time experimental data of each specimen were then fitted by Equation (2), where the only unknown parameter is the permeability of the tissue K. The fitting procedure was implemented by means of a user-defined procedure in (Scilab 5.5.2, ESI Group, France), minimizing the mean square error between numerical results and experimental data.
(3)$$\left[I\right]=\frac{1}{N}\sum_{i=1}^{N}[1-\frac{H\left(t_{i}\right)}{H_{i,exp}}{]}^{2}$$
where *N* is the number of experimental data acquired, *H_i,exp_* the experimental value obtained at time *t_i_,* and *H(t_i_)* is its estimation through Equation (2). Each sample was then characterized by the specific value of permeability found from the fitting.

Before to proceed with the measurement of the permeability of native, recellularized, and decellularized tissues, experimental device and fitting method were tested on samples of a rough porous polymeric material, in order to estimate the reliability of the proposed approach. In detail, we repeated the permeability measurement on the same sample five times. The permeability measurement was then taken on 8 different samples of the same material.

### 4.12. Tissues Thickness Measurement

Thickness in recellularized tissue sections, previously stained with DAPI to highlight the presence of cells, was measured with the ruler function using ImageJ software (National Institutes of Health, Bethesda, MD, USA, https://imagej.nih.gov/ij/, 1997–2018). For each section, three measurements were taken.

### 4.13. Statistical Analysis

All graphs and statistical analyses were performed using the GraphPad Prism Software 6. Data were reported as means ± standard error. For the comparison of coupled experimental groups, the two-sided Student’s *t*-tests (for parametric dataset) and Mann–Whitney test (for non-parametric dataset) were used. One-way ANOVA with Bonferroni’s post-test (for parametric dataset) and Kruskal–Wallis test with Dunns post-test (for non-parametric dataset) were performed for multiple comparisons. A *p*-value < 0.05 was considered statistically significant (* *p*-value < 0.05; ** *p*-value < 0.01; *** *p*-value < 0.001).

## 5. Conclusions

In conclusion, CRC patient-derived 3D model could represent a useful platform for (i) assessing the role of the CRC microenvironment; (ii) supporting cancer cell maintenance and proliferation; (iii) enabling heterogeneity recapitulation of CRC among patients; (iv) bridging the gap between standard in vitro (2D) and in vivo chemosensitivity assays. From the preclinical point of view, this system would be suitable for assessing new prognostic drug screening. Thanks to this model, it would be possible to evaluate the efficacy of chemotherapeutic agents on tumor cells seeded in their specific microenvironment.

## Figures and Tables

**Figure 1 cancers-12-00681-f001:**
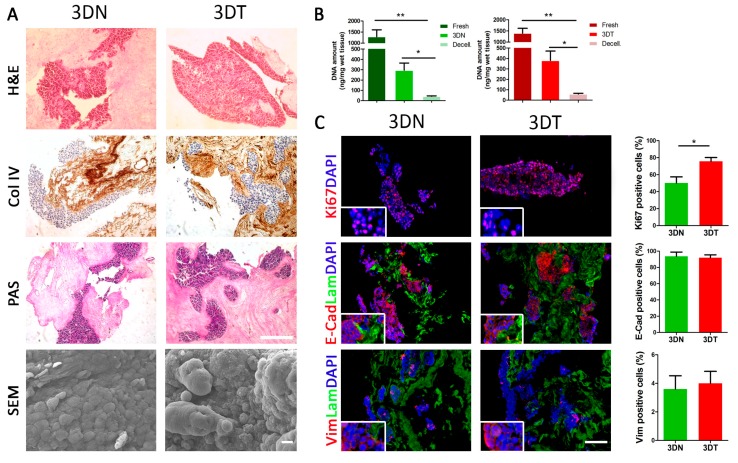
Characterization of matched 3DN and 3DT HT29-recellularized samples. (**A**) histological characterization of sections stained with hematoxylin and eosin (H&E), collagen IV staining (Col IV), and periodic acid-Shiff (PAS) (scale bar = 200 µm). Scanning electron microscopy (SEM) analysis performed on 3DN and 3DT (Scale bar = 10 µm). (**B**) DNA amount quantification in fresh samples, after decellularization and 5 days of culture after seeding of HT29 cells, in both 3DN and 3DT. (**C**) IF staining in 3DT and 3DN and relative quantifications: Ki67, as proliferation marker; E-cadherin, as epithelial marker; vimentin, as mesenchymal marker; laminin to highlight basement membrane; DAPI to counterstain the nuclei (scale bar = 100 µm) (* *p*-value < 0.05; ** *p*-value < 0.01).

**Figure 2 cancers-12-00681-f002:**
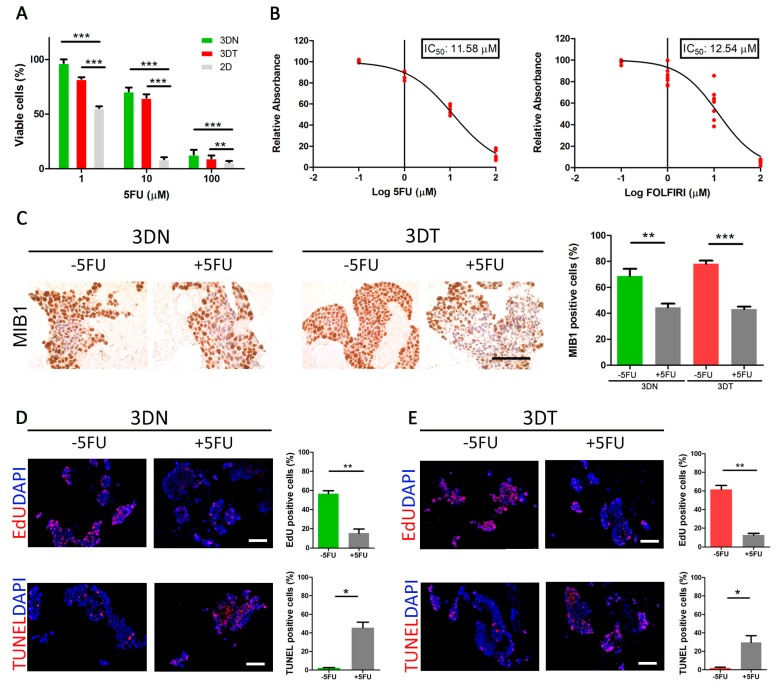
Effect of 5-fluorouracil (5FU) and FOLFIRI ((leucovorin + 5-fluorouracil (5FU) + irinotecan) treatments on HT29-cells cultured in a 3D model. (**A**) Comparison between percentages of viable cells (by absorbance fold-change detection) after administration of 5FU at 1-10-100 μM in 2D cultures and in both 3DN and 3DT models. (**B**) Calculation of 5FU and FOLFIRI 3D IC_50_ by nonlinear regression. (**C**) MIB1 immunohistochemistry before and after administration of 3D-calculated IC_50_ in both 3DN and 3DT; comparison of percentages of MIB1^+^ cells before and after treatment (scale bar = 100 µm). (**D**) EdU staining as a marker of proliferation and TUNEL staining as a marker of apoptosis, before and after 5FU treatment in 3DN and relative quantification (scale bar = 100 µm). (**E**) EdU and TUNEL before and after 5FU treatment in 3DT and relative quantification (scale bar = 100 µm) (* *p*-value < 0.05; ** *p*-value < 0.01; *** *p*-value < 0.001).

**Figure 3 cancers-12-00681-f003:**
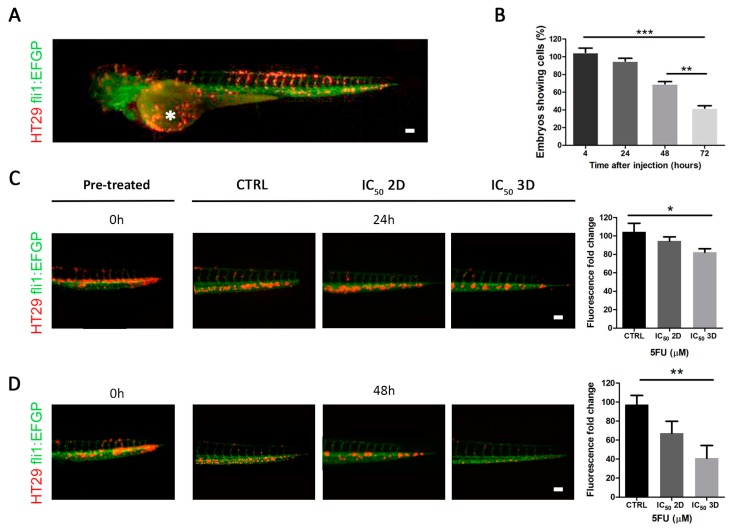
Effect of 5FU treatment on in vivo zebrafish model. (**A**) Tg(fli1:EGFP) zebrafish embryo (with green fluorescent vessels) xenotransplantated with Dil marked HT29 cells (red), injected into the duct of Cuvier (white asterisk). (**B**) The monitoring of zebrafish embryos showed viable cells after 24-48-72 h post-injection. (**C**) Analysis of Dil^+^ HT29 injected cells in zebrafish embryos, pre-treatment (time 0), after 24 h of treatment with DMSO (control group), IC_50_ 2D, IC_50_ 3D, and relative quantifications (scale bar = 100 µm). (**D**) Analysis of Dil^+^ HT29 injected cells in zebrafish embryos, pre-treatment (time 0), after 48 h of treatment with DMSO (control group), IC_50_ 2D, IC_50_ 3D, and relative quantifications (scale bar = 100 µm) (* *p*-value < 0.05; ** *p*-value < 0.01; *** *p*-value < 0.001).

**Figure 4 cancers-12-00681-f004:**
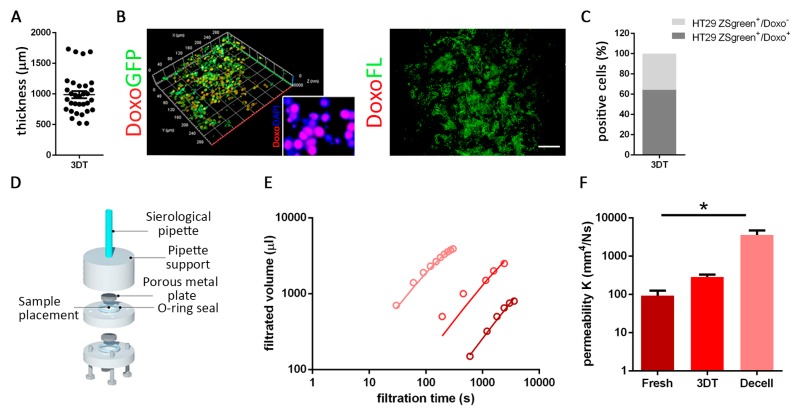
Drug diffusion evaluation. (**A**) Thickness (µm) of a 3DT. (**B**) Doxorubicin (doxo; red) diffusion assay in 3DT, repopulated with HT29 ZSgreen positive cells (green). In the enlargement, the same fixed samples were evaluated with immunofluorescence to underline the co-localization between drug (doxo) and nuclei (DAPI). Absence of non-specific binding between doxo and extracellular matrix (ECM) (scale bar = 100 µm). (**C**) Quantification of co-localized signals. (**D**) In-house developed permeability device. (**E**) Experimental results of the filtration process carried out on the tumor colon fresh, 3DT, and decellularized (decell). Open circles refer to the experimental values, and solid black lines to the Equation (2) fitted to experimental data, and the charts report the estimated values of permeability K (mm4/Ns). (**F**) Quantification of permeability measurement obtained from fresh, 3DT, and decellularized (decell) tumor samples. (* *p*-value < 0.05; ** *p*-value < 0.01; *** *p*-value < 0.001).

**Table 1 cancers-12-00681-t001:** Clinical and pathological characteristics of the 23 patients enrolled in the study. TNM: Classification of Malignant Tumors.

Age	Media	67.73
	Interval	39–83
Sex	male	15 (65.2%)
	female	8 (34.5%)
Grading	G1	2 (8.7%)
	G2	16 (69.6%)
	G3	5 (21.8%)
Staging (p-TNM)	I	1 (4.3%)
	II	6 (26%)
	III	10 (43.4%)
	IV	6 (26%)
Type of surgery	sigmoidectomy	9 (39.1%)
	hemicolectomy dx	9 (39.1%)
	hemicolectomy sx	5 (21.8%)

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
