# Peer review of "Recellularized Colorectal Cancer Patient-Derived Scaffolds as In Vitro Pre-Clinical 3D Model for Drug Screening"

_cancers, 2020, doi:10.3390/cancers12030681_

Round 1
Reviewer 1 Report
The authors have taken into account all the comments. The paper is publishable on Cancers
Author Response
No comments and suggestions have been received from the Reviewer
Reviewer 2 Report
The authors properly revised their manuscript according to reviewers' comments. So, I recommend to publish as it is.
Author Response

(The authors gave the same response as above.)

Reviewer 3 Report
I appreciate the authors put much effort to fix the issues that were mentioned in the original review. The corrections implemented by the Authors improve the quality of the manuscript, however there are still some weakness that should be fixed before manuscript could be recommended for publication. The Authors performed additional experiments and calculated IC50 in tumor cells in conventional 2D and 3DT conditions using FOLFRI. In my opinion, the obtained results increase the substantiveness of the authors' work and should be included in the main part of the manuscript. This will confirm the statement contained in the title of the work about the usefulness of the pre-clinical 3D model for drug screening.
Author Response
Please see the attachment.

This manuscript is a resubmission of an earlier submission. The following is a list of the peer review reports and author responses from that submission.
Round 1
Reviewer 1 Report
Authors proposed an interesting idea a patient-derived 3D preclinical model for testing anti-cancer drugs. The main strength of the manuscript is providing a batter platform to study the effectiveness of drug action that can more accurately mimic drug sensitivity in vivo than a 2D cellular model. However, manuscript have some weakness that should be improved before manuscript could be recommended for publication:
Major points:
1.) I must to admit the biggest problem of the manuscript is showing that the model proposed by the authors works on the basis of only one drug (5FU) used in the treatment of colorectal cancer. I strongly recommend repeating key experiments using at least one other drug, e.g. irinotecan, platinum-based drug or any other relevant.
2.) Secondly, in the first part of the work, the authors show the results of research on two types of 3D models: derived from normal muscoca (N) and cancer lesion (T) from CRC patients. For unknown reasons, this topic is not discussed later in the work and in the discussion section. Do any of the models have an advantage over another? Alternatively, there are no differences and can the proposed 3D model be developed with tissue from healthy donors?
3). I also did not find an explanation as to what criteria the authors used in the choice of cell lines: HT-29 and HCT116. If the Authors would like to emphasize the differences between the lines, I would suggest choosing instead of HCT116 those that have more invasive phenotype, e.g. LoVo or Caco-2.
Minor points:
1). Lines 144-145: whether the observed changes are statistically significant?
2). The description of Figure 2 lacks information which cell line is shown in the figure.
3). For which model, 3DN or 3DT the cytotoxicity plot was shown and the IC50 value was calculated in Figure 2B and Supplementary figure 1D?
4). Do Ki67 and MIB1 refer to the same protein? If so, please unify the names in the entire manuscript. If not, please include anti-MIB1 antibody within the materials and methods section.
5). IC50 value calculated for which model (3DN or 3DT) was included in the study of 5FU treatment on in vivo in a zebrafish model?
6). Lines 307-308: Authors mentioned about E-cadherin and vimentin expression in both 2D and 3D condition. I could not find such results for 2D model.
7). Please provide detailed information about zebrafish treatment with 5FU in Materials and Methods section.
Reviewer 2 Report
In this manuscript, the authors examined whether decellularized ECM (dECM) from normal and cancerous colorectal tissues could be used for pharmacological study of anti-cancer drugs. The authors demonstrated that dECM increased 5FU resistance in dECM and concluded that their dECM is suitable for pharmacological study of anti-cancer drugs. Overall, the aim is important for the researchers in pharmacology, cancer biology, and biomaterial engineering. And some of results are interesting. However, the results did not support many of their claims. Therefore, I cannot recommend to publish this manuscript at present form. Specific comments are below.
1) In Figure 1C, the authors demonstrated the results of Ki67 staining. The graph showed that Ki67-positive cells were greater in 3DT than 3DN. However, it seemed that more Ki67-positive cells were observed in 3DN than 3DT. The authors should show more suitable photos.
2) In section 2.2., the authors claimed “IF analysis evidenced a decreased proliferation rate in the 3D model when compared with the cells grown in conventional 2D condition”. However, there are no 2D results. I could not check whether the authors’ claims were true. The authors should show the results oin 2D culture of both HT29 and HCT116 cells.
3) In Figure 2B, the authors calculated IC50 in 3D culture. Which dECM did the authors used for the culture?
4) Additionally, the authors should compare IC50 in 3DT and 3DN. It is clinically known that the chemoresistance increased according to cancer progression. Are there any differences between 3DN and 3DT? If so, please discuss.
5) In Figure 2D and 2E, the authors showed only the results of 3DT. Please also show the results of 3DN.
6) In Figure 4A, the authors demonstrated the diffusion of doxorubicin into reconstructed cancer tissues with the dECM. However, it seems that the drugs can be easily diffused into the center of reconstructed tissues. The authors should show the size (including the thickness) of the dECM.
7) In section 2.5., the authors described “confirmed the accessibility of the cells seeded in the scaffold to the drug administered through the culture media, even in the most inner areas of the construct. These data indicated that our 3D model is suitable for in vitro drug testing”. For drug testing in vitro, too high diffusion of drugs is problems in my knowledge. According to the authors’ result (Figure 4E-G), the permeability of dECM was higher than fresh tissue. Why could the authors conclude that their dECM is suitable for drug testing? Additionally, the authors should compare the permeability between fresh tissue and reconstructed tissues with the cells and dECM.
8) In Discussion section, the authors claimed “HT29 cells used to recellularized the scaffolds maintained the expected “epithelial phenotype”, showing an high expression of E-cadherin coupled with a low expression of vimentin (line 305-306). However, there are no control of E-cadherin and vimentin in Figure 1C. How did the authors determine these expression levels as high and low, respectively?
Reviewer 3 Report
The manuscript by Sensi, Agostini et al. entitled “Recellularized colorectal cancer patient-derived scaffolds as in vitro pre-clinical 3D model for drug screening” presents the application of colorectal cancer patient-derived scaffolds to test the cytotoxic activity of drug models. Despite the work is well structured, needs a major revision. The recellularized CRC scaffold can represent an interesting model for the cancer community but is unclear how can be reproduced the tumor vasculature. It is well known, that the tumor microenvironment is a complex system, including the surrounding blood vessels, immune cells, fibroblasts, signaling molecules and the extracellular matrix (ECM). The authors in this work are using only few of these. If the manuscript will be revised by taking the following comments into account, reviewer would be willing to recommend this paper for publication in Cancers. The comments were listed as below.
Introduction section: shows a serious of issues: 1) the part (pag 1, lines 43-65), describing the CRC disease, is too long presenting detailed information on CRC unnecessary which could be summarized in a few lines. More information in the introduction should be dedicated to the function of the extracellular matrix on cancer in vitro models 2) relevant/recent works on recellularized colorectal cancer scaffolds are not cited and discussed (i.e. Tian et al., Nat. Bio Eng. vol. 2, pag. 443–452(2018); Landberg et al., Patient-derived scaffolds uncover breast cancer promoting properties of the microenvironment, Biomaterials, vol. 235, 11970 (2019); Rijal et al. Science Advances Sep 2017:Vol. 3, DOI: 10.1126/sciadv.1700764). What is the advantage/improvement of the proposed work, compared to the aforementioned literature? It is well known that collagen, laminin and matrigel/geltrex are also used to provide a 3D culture matrix, what is the advantage to use patient-derived scaffolds? The claim/novelty of the work must be reinforced and explained, clearly.Figure 1: the authors characterized the recellularized scaffold but is missed imaging of decellularized scaffold. SEM characterization, hematoxylin and eosin staining of decellularized patient derived scaffold should be added in a separate panel before figure 1. The DNA amount of decellularized scaffolds in both healthy and tumor tissues was correctly quantified, but imaging characterization is needed. Macrophotographs of the fresh (cellularized) and processed (decellularized) scaffold, in a representative panel, showing the typical color change from red to white it is highly recommended. Minor drawbacks of Figure 1: what magnification was used to histologically characterize the sections stained with Haematoxylin and Eosin (H&E) in figure 1A? It is a 20x magnification?
In Figure 3 the authors investigated the effect of 5FU treatment on in vivo zebrafish model. Why the HT29 cells, immediately after injection into the duct of Cuvier, are distributed along all the parts of embryos? If injected in the duct, the zebrafish should display the accumulation of the tumor cells in the injection site, and therefore the migration towards the caudal zone. The authors should clarify this aspect in the manuscript.
In Figure 4 is reported the characterization of fresh and decell scaffold but is unclear why the authors inserted these pictures here. The characterization of the decellularized scaffold, as suggested before, should be inserted in a separate panel before figure 1. In addition, Figure 4E is a blurred scheme with a text too small. The reviewer suggests to re-organize the figure 4 without figures 4D.
To improve this manuscript, the discussion section should be revised especially explaining future development. In fact, the authors used only tumor cells, that are not enough to mimic the tumor environment. Fibroblasts, immune cells and leaky vasculature are others important components. How to integrate these components? Is it possible? The scaffold can be integrated in some device (i.e. microfluidic chip) to reproduce the blood circulation making possible experiment in flow?
Minor drawbacks: There are some typos and grammatical errors. For example in the Materials and Methods section, paragraph 4.8. “Drug treatment and cytotoxicity assay” : “cells per well in 96-well plates and treated with different concentrations of 5FU from 0.1 μM to 1000 μM for 72 hours”. The drug concentration range is 0.1-100 µM. Authors should check the manuscript carefully.
